# Proteomic Analysis of Low-Grade, Early-Stage Endometrial Carcinoma Reveals New Dysregulated Pathways Associated with Cell Death and Cell Signaling

**DOI:** 10.3390/cancers13040794

**Published:** 2021-02-14

**Authors:** Álvaro López-Janeiro, Ignacio Ruz-Caracuel, Jorge L. Ramón-Patino, Vivian De Los Ríos, María Villalba Esparza, Alberto Berjón, Laura Yébenes, Alicia Hernández, Ivan Masetto, Ece Kadioglu, Virginie Goubert, Victoria Heredia-Soto, Rodrigo Barderas, José Ignacio Casal, Carlos E. de Andrea, Andrés Redondo, Marta Mendiola, Alberto Peláez-García, David Hardisson

**Affiliations:** 1Department of Pathology, Hospital Universitario La Paz, IdiPAZ, 28046 Madrid, Spain; aljaneiro@salud.madrid.org (Á.L.-J.); ignacio.ruz@salud.madrid.org (I.R.-C.); alberto.berjon@salud.madrid.org (A.B.); laura.yebenes@salud.madrid.org (L.Y.); 2Department of Medical Oncology, Hospital Universitario La Paz, IdiPAZ, 28046 Madrid, Spain; jorge.ramon@hospitalreyjuancarlos.es (J.L.R.-P.); andres.redondo@salud.madrid.org (A.R.); 3Proteomics Facility, Centro de Investigaciones Biológicas (CIB-CSIC), 28039 Madrid, Spain; vrios@cib.csic.es; 4Department of Pathology, Clínica Universidad de Navarra, University of Navarra, 31008 Pamplona, Spain; mvillalbae@unav.es (M.V.E.); ceandrea@unav.es (C.E.d.A.); 5Center for Biomedical Research in the Cancer Network (Centro de Investigación Biomédica en Red de Cáncer, CIBERONC), Instituto de Salud Carlos III, 28046 Madrid, Spain; victoriam.heredia@salud.madrid.org (V.H.-S.); marta.mendiola@salud.madrid.org (M.M.); 6Molecular Pathology and Therapeutic Targets Group, La Paz University Hospital (IdiPAZ), 28046 Madrid, Spain; 7Department of Obstetrics & Gynecology, Hospital Universitario La Paz, IdiPAZ, 28046 Madrid, Spain; ahernandez@salud.madrid.org; 8Faculty of Medicine, Universidad Autónoma de Madrid, 28029 Madrid, Spain; 9Akoya Biosciences, 100 Campus Drive, 6th Floor, Marlborough, MA 01752, USA; imasetto@akoyabio.com (I.M.); vgoubert@akoyabio.com (V.G.); 10Lunaphore Technologies SA, Route de Lully 5C, CH-1131 Tolochenaz, Switzerland; ece.kadioglu@lunaphore.com; 11Translational Oncology Research Laboratory, Hospital La Paz Institute for Health Research (IdiPAZ), 28046 Madrid, Spain; 12Chronic Disease Programme, UFIEC, Instituto de Salud Carlos III, 28220 Majadahonda, Madrid, Spain; r.barderasm@isciii.es; 13Centro de Investigaciones Biológicas (CIB-CSIC), Department of Molecular Biomedicine, 28039 Madrid, Spain; icasal@cib.csic.es

**Keywords:** endometrial cancer, low grade, proteomics, pathways, SLIT/ROBO, necroptosis, ferroptosis, immune microenvironment

## Abstract

**Simple Summary:**

Low-grade, early-stage endometrial cancer (EC) is the most frequent malignant tumor of the uterine corpus. Our study aimed to assess dysregulated pathways in this specific subset of EC through proteomic analysis. We describe and validate the dysregulation of the SLIT/ROBO signaling pathway, as well as cellular death processes such as necroptosis and ferroptosis. We identify several immune-related pathways, with a dominance of innate immune response associated pathways. Our findings reveal the singular biology of low-grade, early-stage ECs and could guide future research in the field.

**Abstract:**

Low-grade, early-stage endometrial carcinoma (EC) is the most frequent malignant tumor of the uterine corpus. However, the molecular alterations that underlie these tumors are far from being fully understood. The purpose of this study is to describe dysregulated molecular pathways from EC patients. Sixteen samples of tumor tissue and paired healthy controls were collected and both were subjected to mass spectrometry (MS)/MS proteomic analysis. Gene ontology and pathway analysis was performed to discover dysregulated pathways and/or proteins using different databases and bioinformatic tools. Dysregulated pathways were cross-validated in an independent external cohort. Cell signaling, immune response, and cell death-associated pathways were robustly identified. The SLIT/ROBO signaling pathway demonstrated dysregulation at the proteomic and transcriptomic level. Necroptosis and ferroptosis were cell death-associated processes aberrantly regulated, in addition to apoptosis. Immune response-associated pathways showed a dominance of innate immune responses. Tumor immune infiltrates measured by immunofluorescence demonstrated diverse lymphoid and myeloid populations. Our results suggest a role of SLIT/ROBO, necroptosis, and ferroptosis, as well as a prominent role of innate immune response in low-grade, early-stage EC. These results could guide future research in this group of tumors.

## 1. Introduction

Endometrial carcinoma (EC) is the most frequently diagnosed gynecologic malignancy in the European Union, with more than 73,000 new cases estimated to have been diagnosed in 2020 [1]. Most patients with EC are diagnosed at early stages of their disease (80% in stage I or II) and present as low-grade endometrioid endometrial carcinomas (EEC) [2]. Surgery is the primary treatment for EC, followed by adjuvant radiotherapy or chemotherapy depending on risk group stratification, which is mainly based on histology (endometrioid vs non-endometrioid), tumor grade (grade 1–2 vs grade 3), depth of myometrial invasion (<50% vs ≥50%), and the presence or absence of lymphovascular space invasion [3,4]. Although most patients have a favorable outcome, around 10% of them will progress with local or distant relapses [2]. To date, systemic therapy has limited benefits in this subset of patients, and its role is poorly defined in the adjuvant setting or after relapse [5].

A new molecular classification of EC has been proposed based on the genomic features of the tumors [6]. Genetic alterations, such as *POLE* mutation, copy number alterations, and microsatellite instability, can be used to classify EC and guide post-surgical adjuvant treatment for women with more aggressive tumors. Proteomic analysis of EC has also been used to identify potential diagnostic markers using endometrial tissue and uterine lavage samples [7,8,9]. In addition, studies have revealed novel associations between protein expression and the genomic profiles of tumors [10]. Most of these studies included the endometroid and serous subtypes of EC, revealing major molecular differences between these tumors.

Despite much progress in the understanding of endometrial tumor biology, there is still a considerable lack of knowledge in the pathways that govern tumorigenesis of low-grade, early-stage EEC. Filling this gap could enable the development of specific therapies for different subsets of endometrial tumors. Little has been published about the specific proteomic profile of low-grade, early-stage ECs, with most studies focusing on isolated protein dysregulation [11,12,13] or the usefulness of proteomics in endometrial cancer diagnosis [14].

In the present study, we aimed to characterize low-grade, early-stage EEC through quantitative comparative analysis of the endometrial proteome from formalin-fixed paraffin-embedded (FFPE) samples. We focus on previously undescribed pathways in the pathogeneses of EC which are dysregulated in EEC, and seem relevant for tumor biology. These new findings could help to guide future research into therapeutic alternatives for these patients.

## 2. Results

The proteomics study comprised a discovery and a validation cohort. The discovery set consisted of sixteen low-grade, early-stage EC formalin-fixed, paraffin-embedded (FFPE) tissue samples with paired healthy controls. As a validation cohort, proteomic data from 11 tumor samples and 6 healthy controls were obtained from the CPTAC study [10]. An outline of our experimental design is depicted in Figure 1. Overall, 3112 proteins were identified in the discovery set, and 9802 in the validation set with a global false-discovery rate (FDR) below 1% (Appendix A). Initially, as an explorative approach, principal component analysis (PCA) and hierarchical clustering was carried out with the discovery and validation set. Tumor and healthy tissue were clearly clustered according to their proteomic protein expression profiles (Figure 2a and Appendix A). Using an FDR threshold of 5%, 572 (18.3%) proteins were differentially expressed in the discovery set, which resulted in 854 identified pathways. In the validation set, there were 7775 (78%) differentially expressed proteins that yielded 5856 pathways (Figure 2a). After matching pathways identified in both cohorts, a total of 503 pathways were cross-validated in the discovery and validation cohort. The mean number of proteins per pathway identified was 14.34 and 106.32 for the discovery set and the validation set, respectively. 

Additionally, the mean percentage of genes from each pathway identified in the discovery and validation set was 11.56% and 51.86%, respectively. 

Most of the cross-validated pathways were related to essential biological cell functions, including cell metabolism, nucleic acid synthesis, and protein translation. Importantly, pathways classically associated with the pathogenesis of EEC and general carcinogenesis were robustly identified in our analysis, including MAPK signaling, DNA repair processes, PI3K-Akt dysregulation, and steroid hormone receptor-associated pathways. These served as positive internal control of our pathway discovery pipeline (Figure 2b). 

We screened cross-validated pathways and selected the pathways related to cell signaling, immune response and cell death to be subjected to further study. WNT pathways and L1CAM interaction pathways were altered according to our proteomic analysis. We orthogonally validated the WNT pathway by measuring the proportion of β-catenin nuclear stain using immunohistochemistry (IHC) in a cohort of 265 patients with low-grade, early-stage EEC. Out of 245 tumors evaluable for β-catenin status, 26 (10.6%, 95% confidence interval = 7.05–15.16%) showed nuclear β-catenin expression (Figure 3b). Furthermore, our proteomic analysis found β-catenin protein to be up-regulated in both the discovery and validation sets (Appendix A). Moreover, HMGB3 (a chromatin binding protein involved in the regulation of cell cycle through several mechanisms, including WNT pathway) [15,16], was strongly up-regulated in our proteomic analysis, being the third most up-regulated protein. A similar pattern was found in the validation cohort (Appendix A). To further investigate this finding, we measured the mRNA levels of *HMGB3* in tumor and paired healthy tissue. In line with our proteomic findings, the mRNA levels of *HMBG3* were up-regulated in the tumor tissue. Although results were not statistically significant when analyzed as independent cohorts, most paired samples (7 out of 10) showed increased HMGB3 expression at the intra-subject level (Figure 3c). In order to ascertain the presence of L1CAM in EEC, we measured the expression of this protein by IHC, revealing membranous positivity in 13 out of 213 tumors analyzed (6.1%, 95% confidence interval = 3.29–10.21%) (Figure 3b). The SLIT/ROBO signaling pathway was one of the most robustly identified dysregulated pathways, with more than 20% of pathway-related proteins and more than 25 proteins identified in the discovery cohort (Figure 2b and Appendix A). Similar results were obtained in the validation set. To further validate our proteomic findings, we measured *SLIT2* and *ROBO2* mRNA levels, which are the main ligand and receptor of the SLIT/ROBO signaling pathways, respectively, and are widely expressed outside the central nervous system. In line with our proteomic findings, results of the mRNA analysis showed up-regulation of the *ROBO2* receptor with concomitant down-regulation of the *SLIT2* ligand in tumor tissue compared to healthy paired tissue (Figure 3d). In addition, this transcriptomic analysis was consistent with individual protein measures in the validation cohort, also showing *SLIT2* down-regulation with concurrent *ROBO2* up-regulation (Figure 3f). Furthermore, all SLIT/ROBO pathway-associated proteins identified in the discovery cohort showed a pattern of up-regulation in tumor tissue (Appendix A). 

Additionally, several pathways associated with cell death were detected in our proteomic analysis. Aside from apoptosis related pathways, which are frequently dysregulated in malignancies, we identified necroptosis and ferroptosis as potential cell death pathways altered in low-grade, early-stage EEC (Appendix A). We further explored this finding, analyzing *CFLAR* mRNA levels in low-grade, early-stage EEC and healthy paired tissue. *CFLAR* encodes cFLIP, a protein which acts as a negative regulator of necroptosis. As expected, *CFLAR* levels were down-regulated in the tumors (Figure 3e). Furthermore, necroptosis-associated effector proteins RIPK1, RIPK3 and MLKL, identified in the Clinical Proteomic Tumor Analysis Consortium (CPTAC) validation cohort, were all up-regulated in the tumors, in line with our transcriptomic findings (Figure 3f). Assessment of the expression pattern of key ferroptosis regulatory proteins obtained from the proteomic data of the LAPAZ and CPTAC cohorts provided additional evidence supporting pathway dysregulation. Proteins associated with ferroptosis suppression, such as the glutathione-dependent lipid antioxidant (GPX4) and the glutathione synthetase (GSS), were both up-regulated in tumors compared to healthy tissue. This pattern of expression was also found for these individual proteins in the discovery set proteomic analysis (Appendix A). In addition, the newly characterized ferroptosis suppressor AIFM2 (also known as ferroptosis suppressor protein 1, FSP1) was also up-regulated in the tumors. Moreover, the transferring receptor 1 protein (TFRC), involved in the homeostasis of intracellular free iron, a key factor triggering ferroptosis, was also up-regulated in tumor tissue. These findings support the existence of alternative cell death pathways other than apoptosis in low-grade, early-stage EEC. These key pathway proteins were also differentially expressed after correcting for multiple comparisons (Appendix A). Immune response-related pathways identified by our proteomic analysis were consistent with diverse immune cell infiltrates, including a variety of cytokines and immune cell-related pathways. Among these, the most robustly identified pathways were associated with innate immune response, including neutrophil activation and myeloid differentiation. Routes related to adaptive immune response such as TCR signaling, MHC class II antigen presentation, and INF-gamma were also detected, albeit with variable proportion between discovery and validation cohorts. In order to validate these results, we measured the immune densities of CD8 positive cytotoxic T cell and CD68 positive myeloid cell populations in 15 cases of healthy uterine tissue. Both cell populations were scarce, with median values of CD8 and CD68 cells per mm^2^ of 14.1 and 0, respectively (Figure 4a).

Then, multiplexed quantitative analysis was used to determine the CD8+ T cell and CD68 densities in low-grade, early-stage EEC using very well characterized and clinically annotated tissue microarrays (TMAs) (see Methods). The analysis of 230 spots from 162 patients revealed a significantly higher infiltration of CD68+ macrophages and CD8 T cells compared to healthy uterine tissue. Overall, the median CD68 and CD8 positive cells per mm^2^ was 77.72 and 96.37, respectively. The immune cell infiltrates were more prominent in the tumor stroma than in the tumor epithelium (Figure 4a,b). Additionally, we appreciated high heterogeneity in the ratio of CD68 to CD8 cells (Appendix A). Overall, the median CD68/CD8 Log2 ratio was –0.64, revealing a dominance of CD8 infiltrates. However, interquartile range revealed great variability (2.37), with p25 and 75 being –1.68 and 0.68, respectively. In an effort to determine the association between the immune cell densities of lymphoid and myeloid cell markers, we performed Spearman correlation analysis. We found that CD8+ T cell densities were moderately correlated with the myeloid cell marker CD68 (Spearman’s rank-order correlation; *r* = 0.38; *p* < 0.001) (Appendix A). 

## 3. Discussion

In the present study, we have focused on the proteomic differences between low-grade, early-stage EEC and their healthy tissue counterpart. Previous studies have provided relevant insight into the molecular features of EC, revealing different molecular subtypes that correlate with clinical outcomes [6]. Proteogenomic analyses of endometrial tumors have yielded interesting correlations between driver mutations, protein expression, and histone methylation profiles, and have also revealed immune response heterogeneity among tumors [10]. However, none of these studies have focused on the specific subtypes of endometrial carcinoma, mostly reflecting differences between biologically divergent neoplastic processes. Our analysis of low-grade, early-stage EEC identified pathways classically associated with endometroid tumors, such as PI3K-Akt and WNT pathways. Our data support that up-regulation of these pathways plays a role in low-grade, early-stage EEC. This is also in line with previous studies analyzing driver mutations in normal endometrial glands, which described alterations in mitogen-activated kinases [17].

Interestingly, we have identified the SLIT/ROBO signaling pathway to be robustly dysregulated in these tumors. Molecules acting in this pathway were first discovered as regulators of axon growth [18]. They have been linked to cancer-associated processes such as angiogenesis, cell migration, metastasis, and the regulation of leucocyte chemotaxis [19]. Interestingly, Pan-SLIT and ROBO1 immunohistochemical expression in EC has been linked to increased recurrence rates [20]. In addition, SLIT2 deficient signaling has been associated with a worse prognosis and increased risk of hematologic malignancy progression [21]. In this sense, point mutations, somatic copy number variations, and changes in gene promoter methylation status have been described in genes associated with these pathways in other malignancies, such as myelodysplastic syndromes and pancreatic cancer [21,22]. Importantly, other studies have identified similar patterns of ligand–receptor dysregulation, with decreased SLIT2 and increased ROBO in tumors compared to healthy counterparts [23]. Moreover, selective deletion of *SLIT2* in tumor cells in a murine tumor model caused increased tumor metastatic potential [24]. In addition, the SLIT-2/ROBO axis has been found to regulate β-catenin function, which may link SLIT/ROBO signaling to WNT pathway dysregulation, which is commonly altered in EEC [23]. Our results warrant further studies on the potential role of SLIT/ROBO as a therapeutic target and biomarker for the prognosis of low-grade EEC.

In the present study, we found pathways classically associated with low-grade EEC oncogenesis. β-catenin (*CTNNB1*), a key mediator of WNT pathway activation, was mutated in a subset of low-grade EEC. In line with previous studies, we found that WNT pathway activation is a feature of these tumors, where it is mediated by β-catenin nuclear translocation [25]. The fact that low prevalence of nuclear β-catenin was detected in our study has to be evaluated, considering the limited sensitivity of TMAs to score focally expressed proteins. In addition, we report novel mechanisms that might regulate WNT pathway activation in low-grade EEC, namely the SLIT/ROBO pathway (see above) and HMGB3 overexpression. Interestingly, other groups have reported an association between HMGB3 expression and adverse outcomes in all-type EC [26]. However, the prognostic role of HMBG3 in the specific subset of low-grade, early-stage EEC is still to be elucidated.

Evidence suggests that L1CAM expression is a marker of worse prognosis in endometrial cancer. L1CAM expression is commonly observed in high-grade endometrial tumors (including serous and clear cell carcinomas) and is associated with tumor vascular invasion [27]. We have consistently identified L1CAM pathway activation by proteomics and IHC in our series of low-grade, early-stage EEC. Despite having detected L1CAM in a small subset of early-stage EEC, previous reports have associated the expression of L1CAM protein in low-grade tumors with worse overall survival [28], which warrants further study of this pathway in the biology of low-grade EEC.

Regulated cell death, a hallmark of cellular homeostasis, is frequently altered in neoplastic processes. Attention has been drawn to the diversity of cell death pathways in health and disease; some of these cell-death programs may impair therapeutic approaches or be exploitable for treatment purposes [29,30]. Of these, apoptosis and necroptosis were the cell death-associated pathways more robustly identified in our study. Necroptosis is a caspase-independent cell death mechanism that can be triggered by several stimuli, including pattern recognition receptors, TNF signaling, metabolic stress, and chemotherapeutic agents. The negative regulation of necroptosis has been associated with variable clinical outcomes in patients with different cancer types [31]. The identification of necroptosis as a type of cell death in low-grade, early-stage EEC could enable future therapeutic strategies for these patients. Additionally, ferroptosis was also dysregulated in our analysis. This type of cell death is intimately associated with cell oxidative stress, derived from increased metabolic activity. Notably, *KRAS* and *BRAF* mutant cell lines, as well as *EGFR*-expressing cell lines, show increased susceptibility to ferroptosis [30]; we hypothesize that some of the active oncogenic pathways in EC may also increase the cell oxidative stress. Furthermore, reports suggest that peroxisomal activity in cancer cells may produce lipid intermediates that act as substrates for ferroptosis [32]. Our results suggest that this increased susceptibility to ferroptosis is quenched by the up-regulation of inhibitors such as GPX4, AIFM2 (FSP1) and GSS, which could potentially be targeted in low-grade EEC [33,34]. Further supporting our results, increased *MLKL* and *AIFM2* mRNA expression is associated with a favorable prognosis in EC, according to the Human Protein Atlas [35].

Furthermore, necroptosis and ferroptosis are linked with immune response, because these cell death processes release damage-associated molecular patterns (DAMPS) that can be sensed by immune cells to amplify inflammatory responses. With the advent of immunotherapy, there is increased awareness of the impact of immune microenvironment in cancer behavior and clinical outcomes. Tumor-immune microenvironments differ across EC subtypes, and show spatial and temporal heterogeneity [36]. Adaptative immunity against tumor neo-antigens is supposed to drive sustained responses with current immunotherapies. However, the fact that most patients do not benefit from these therapies suggests that other immune cell populations may play major roles in the tumor microenvironment. In the present study, aside from pathways associated with adaptative immunity, we have robustly identified pathways related to innate immune activation. Our results are in line with animal models and previous analyses of immune infiltrates in EC that have demonstrated increased neutrophil infiltration in these tumors [37,38]. The role of innate immune response in cancer biology is a field of growing interest, with contemporary studies suggesting a central role of macrophages in regulating anti-tumoral immune response and the discovery of neutrophil recruitment by tumors to promote growth and metastases [39,40,41]. On the other hand, some reports have demonstrated the down-regulation of class 1 MHC molecule expression in EEC, as well as reduced CD8 expression in cytotoxic tumor-infiltrating T cells, which could limit antigen presentation and adaptative immune response [42,43]. These findings could partially explain the dominance of innate immune response pathways over lymphocyte-associated pathways in our study. Our results highlight the importance of innate immune cells in low-grade EEC, which may guide future immunotherapy strategies in these specific tumor subtypes beyond current approaches based on the PD1–PDL1 axis blockade [44].

Our study has several limitations that should be acknowledged. Firstly, our proteomic pipeline was based on a limited number of FFPE tissue samples. Using an FFPE sample as the protein source reduces the potential number of proteins identified. It is, therefore, possible that relevant pathways could be undetected in this study. In addition, due to the reduced number of patients included in the proteomic analysis, and the excellent outcome of most of these patients, the clinical relevance of these pathways has not been explored.

## 4. Materials and Methods

### 4.1. Discover and Validation Cohort

As a discovery set (LA PAZ cohort), 16 FFPE tumor samples with paired healthy tissue were selected from our archives. Inclusion criteria included patients with low-grade (grade 1 or grade 2), early-stage (International Federation of Gynecology and Obstetrics, FIGO stages I or II) EEC, harboring a p53 wild-type protein expression pattern. Patients were treatment-naïve. Hematoxylin and eosin (H&E)-stained slides of each case were reviewed by an expert pathologist who selected tumor areas and paired healthy tissue. Appendix A shows the clinicopathologic characteristics of the patients included in the study. 

As a validation set (CPTAC cohort), MS data from healthy tissue and low-grade, early-stage EEC samples were obtained from the CPTAC data portal (date of accession, March 8, 2020). Data were obtained from two TMT-10 experiments that only included tumor and healthy tissue from low-grade, early FIGO stage ECs. Eleven tumor samples were obtained. Of these, 6 had available paired healthy tissue. The experimental protocol for the isolation of proteins in this cohort was carried out from fresh frozen tissue. Raw MS spectra files were downloaded and analyzed as for the discovery set (see Data Analysis below (Section 4.5)).

### 4.2. Protein Extraction and Processing

Tissue samples were sectioned using a microtome (7 µm thick), transferred into 1.5 mL tubes and deparaffinized by incubation with xylol for 30 min. After the addition of methanol, samples were vortexed thoroughly and centrifuged for 2 min (15,000 rpm). The supernatant was removed, and tissue samples were air-dried followed by resuspension in RIPA buffer with protease/phosphatase inhibitors. Samples were boiled and sonicated, followed by heating for 2 h at 80 °C (500 rpm). Samples were then reduced by adding Tris(2-carboxyethyl)phosphine hydrochloride (TCEP) to a final concentration of 10 mM and alkylated with iodoacetamide to a final concentration of 40 mM. Protein extracts were cleaned and digested with the SP3 method, as described previously [45,46]. Briefly, 10 µL of SP3 bead stock (Sera-Mag SpeedBead carboxylate-modified magnetic particles; GE Healthcare Life Sciences, Marlborough, MA, USA) for sample (or a 1:10 protein:bead ratio for sections) and 200 µL acetonitrile (ACN; final concentration of 70%) were added to 200 µL of protein extract and incubated for 10 min (1000 rpm). Tubes were mounted on a magnetic rack; supernatants were removed, and beads were washed twice with 70% ethanol and once with ACN (1 mL each). Beads were resuspended in 200 µL 200 mM HEPES pH 8 and digested overnight with trypsin (1:25 trypsin:protein ratio for sections, PierceTM trypsin protease, Thermo Scientific, Waltham, MA, USA; 37 °C, 500 rpm). The next day, spinning down tubes were mounted on a magnetic rack and the supernatant was recovered to a fresh 1.5 mL tube. 

### 4.3. TMT Labeling

To analyze the discovery set, four TMT experiments were performed. TMT reagents (0.8 mg) were dissolved in acetonitrile (40 μL), of which 20 μL was added to the peptides (50 μg). Peptide quantification was performed by a QuBit fluorometer. Following incubation at room temperature for 1 h (500 rpm), the reaction was quenched with hydroxylamine to a final concentration of 0.3% (*v/v*). TMT-labeled samples were pooled at a 1:1 ratio across all 10 samples. For each experiment, the pooled sample was vacuum centrifuged to near-dryness. Pierce High pH Reversed-Phase Fractionation Kits were used to fractionate TMT-labeled digest samples into eight fractions by an increasing acetonitrile step-gradient elution. Fractions were dried in a vacuum centrifuge and resuspended in 0.1% formic acid prior to LC–MS analysis.

### 4.4. Liquid Chromatography and Tandem Mass Spectrometry

Peptide separations were carried out on an Easy-nLC 1000 nano system (Thermo Scientific). For the analysis, the sample was loaded into a precolumn Acclaim PepMap 100 (Thermo Scientific) and eluted in an RSLC PepMap C18, 50 cm long, 75 µm inner diameter and 2 µm particle size (Thermo Scientific). The mobile phase flow rate was 300 nL/min using 0.1% formic acid in water (solvent A) and 0.1% formic acid and 100% acetonitrile (solvent B). The gradient profile was set as follows: 3–7% solvent B for 5 min, 7–25% solvent B for 95 min, 25–60% solvent B for 14 min, 60–95% solvent B for 1 min, and 95% solvent B for 5 min. Four microliters of each sample were injected. MS analysis was performed using a Q-Exactive mass spectrometer (Thermo Scientific). For ionization, 2000 V of liquid junction voltage and 270 °C capillary temperature was used. The full scan method employed a *m/z* 300–1800 mass selection, an Orbitrap resolution of 70,000 (at *m/z* 200), a target automatic gain control (AGC) value of 3 × 10^6^, and maximum injection times of 100 ms. After the survey scan, the 15 most intense precursor ions were selected for MS/MS fragmentation. Fragmentation was performed with a normalized collision energy of 27 and MS/MS scans were acquired with a starting mass of *m/z* 200, AGC target was 10^5^, resolution of 35,000 (at *m/z* 200), intensity threshold of 8 × 10^3^, isolation window of 1.6 *m/z* units, and the maximum IT was 100 ms. Charge state screening was enabled to reject unassigned, singly charged, and ≥7 protonated ions. A dynamic exclusion time of 30 s was used to discriminate against previously selected ions.

### 4.5. Data Analysis

Mass spectra were searched against all entries from the human UniProt Database (downloaded: August 2019) using MaxQuant (version 1.6.8.0, Max Planck Institute of Biochemistry, Martinsried, Germany) [47,48]. Trypsin/P was specified as a cleavage enzyme, allowing up to two missed cleavages and a mass tolerance of 20 ppm Orbitrap for precursor ions and MS2 fragment ions. The product mass tolerance was set to 0.5 Da, and a maximum of two missed tryptic cleavages were allowed. Carbamidomethylation of cysteine was used as a fixed modification, with the oxidation of methionine and acetylation of lysine residue used as variable modifications. A reverse nonsense version of the original database was generated and used to determine the FDR, which was set to 1% for peptide identifications. Reporter ion intensities were adjusted to correct for the isotopic impurities of the different TMT reagents according to manufacturer specifications. Missing value imputations were carried out from a normal distribution (width: 0.3, downshift: 1.8) using Perseus software. The normalization of reporter ion intensities was carried with an internal reference scaling (IRS) method [49]. The mass spectrometry proteomics data have been deposited to the ProteomeXchange Consortium via the PRIDE [50] partner repository, with the dataset identifier PXD022733 and 10.6019/PXD022733. Principal component analysis (PCA) was performed using the FactoMineR v2.2 R package. Hierarchical clustering was conducted using the Ward. D2 algorithm and taking Euclidean distance as similarity metric. Differential gene expression was carried out using a *t*-test and the non-parametric Wilcoxon test for independent samples. The resultant *p*-values were penalized with a false discovery rate (FDR). A threshold of FDR < 0.05 was considered to include each protein in further analyses. Given that both differential expressions yielded overlapping results (with 94% of labeled proteins by Wilcoxon tests being also significant using the *t*-test), we included proteins described by the non-parametric test in further analyses. To perform pathway enrichment, Gene Ontology, Kegg pathways and Human Reactome pathway databases were interrogated using Cytoscape (v3.7.2) and ClueGo (v2.5.6) [51,52]. Only differentially expressed proteins were included in the pathway analysis pipeline. An FDR corrected pathway identification *p*-value of less than 0.15 in the discovery cohort was required for pathway consideration. After independently analyzing the discovery and validation cohorts, only cross-matched pathways were considered for further rounds of validation. 

Statistical analyses were performed using R (v4.0.1), and plots were generated using the ggplot2 R package (v3.3.1) [53]. Correlations were calculated using the Spearman’s ranked correlation method. To analyze individual protein differences, two-sided *t*-tests with the Welch correction were calculated. 

### 4.6. Real-Time Quantitative RT-PCR

Total RNA samples were isolated from representative FFPE tissue blocks from 10 low-grade, early-stage primary EEC using a QIAGEN RNeasy Mini kit following the manufacturer’s instructions and quantified by spectrophotometry. Quantitative real-time polymerase chain reactions were performed in an Mx3005p (Agilent, Santa Clara, CA, USA) using the SYBR^®^ Green Quantitative RT-qPCR Kit (Merck, Darmstadt, Germany). Expression of target RNAs was normalized using *GADPH* and *PPIA* genes as internal controls [54]. A list of forward and reverse primers for *SLIT2*, *ROBO2*, *HMG-B3* and *CFLAR* can be found in Appendix A. Results were analyzed, quantifying the target mRNA expression in relation to housekeeping gene expression. The Wilcoxon test for independent samples was used to calculate *p*-values.

### 4.7. TMA Construction, Immunohistochemistry and Immunofluorescence

For TMA construction, representative non-necrotic central areas of each tumor were marked on H&E slides. Two representative cores of 1.2 mm were obtained from the selected areas of the paraffin block. The tissue cores were arrayed into a receptor paraffin block using a tissue array (TMA) Workstation (Beecher instruments, Silver Spring, MD, USA), as previously described [55]. A total of 265 patients had available tissue for TMA construction, and 14 TMAs containing tumor tissue cores were constructed. TMA spots were considered adequate for analysis if no detachments or staining artifacts were noted. Any spot not fulfilling quality standards was not considered in further analyses. Complete description of the cohort’s clinical characteristics can be found in a previously published article [55].

Immunohistochemistry (IHC) was performed by the Envision method (Dako-Agilent, Glostrup, Denmark) in an automated Omnis platform (Dako-Agilent) with the following antibodies: beta-catenin (mouse monoclonal, clone β-catenin-1, Ready-to-use, Dako-Agilent), L1CAM (mouse monoclonal, clone G-5, 1:100, Santa Cruz, USA), CD8 (mouse monoclonal, clone C8/144B, Ready-to-use, Dako-Agilent) and CD68 (mouse monoclonal, clone KP-1, Ready-to-use, Dako-Agilent). For beta-catenin, positive cases were considered if any nuclear staining was present in tumor cells in at least one of the cores evaluated. L1CAM membranous staining in >10% of tumor cells in any of the spots was considered positive. Cytoplasmic staining was not considered. Immune infiltrates were quantified separately in tumor and healthy tissue. To quantify immune infiltrates in tumor tissue, a subset of seven TMAs from our low-grade, early-stage EEC cohort were selected. A total of 162 patients providing 230 spots were included in the analysis. Multiplex immunofluorescence (IF) validation workflows and protocols have been implemented, as previously described [56,57]. Briefly, 4 μm sections of FFPE tissue from TMAs were deparaffinized, and antigen retrieval was performed using heat-induced antigen retrieval with low pH (pH 6) or high pH (pH 9) solutions. Samples were stained with primary antibodies targeting Cytokeratin (CK) (pan-CK, clone AE1/AE3, dilution 1:150, Novus Biologicals), CD8 (clone 4B11, dilution 1:150, Biorad), and CD68 (clone PG-M1, dilution 1/100, Dako) followed by TSA visualization with fluorophores Opal 690, Opal 540, Opal 620 (Akoya Biosciences), respectively. All steps were performed on the LabSat Research platform (Lunaphore Technologies, Lausanne, Switzerland). This is an automated tissue-staining instrument for rapid immunostaining which utilizes a microfluidic technology for the rapid and uniform delivery of reagents to tissue samples, as previously described [58]. Each TMA section underwent several sequential rounds of antibody staining. In the last round, nuclei were counterstained with spectral DAPI (Akoya Biosciences), and sections were mounted with Faramount Aqueous Mounting Medium (Dako).

Multiplexed immunofluorescence TMA slides were imaged using the Vectra^®^ Polaris™ Automated Quantitative Pathology Imaging System (Akoya Biosciences, Menlo Park, CA, USA). Tissue imaging, spectral unmixing, and phenotyping were performed using inForm^®^ Tissue Analysis Software (v2.4.8, Akoya Biosciences, Menlo Park, CA, USA). 

Each TMA core was scanned and analyzed. For multi-spectral image analysis, a spectral library from spectral peaks emitted by each fluorophore from single stained slides was generated using InForm software (version 2.4.8, Akoya Biosciences) and used for the spectral unmixing of multispectral images, allowing for the identification of all markers of interest. Auto-fluorescence was determined on an unstained tumor core used in this study cohort. Tissue and cell segmentations were performed using InForm software. Tissue segmentation algorithms based on cytokeratin staining were used for the distinct identification of tumor islands (tumor epithelium), stroma areas, and regions of non-interest. Cell segmentation algorithms based on the expression of different markers, such as the nucleus (DAPI), cytoplasm, or membrane, were used to accurately identify each cell. Cell subpopulations were phenotyped in each TMA core, as: CD8+, CD68+, and CK+. Cells negative for these markers were classified as other cell types. All phenotyping and subsequent quantifications were performed blinded to the sample identity. A representative example of the cell and tissue segmentation process is depicted in Appendix A.

To assess immune infiltrates in healthy tissue, whole slides from 15 FFPE tissue blocks from the non-tumoral tissue included in the proteomic analysis were selected and stained for CD8 and CD68. Slides were reviewed, and three representative spots of 0.6 mm of diameter per slide were chosen to count positive cells. Counting was manually performed at 20× magnification aided by an Olympus BX51 microscope equipped with a DP72 digital camera.

## 5. Conclusions

Using proteomics, we have identified novel dysregulated pathways in low-grade, early-stage EEC. The SLIT-ROBO signaling pathway, which is important in tumor biology, seems to play an important role in these tumors. We also found that the activation of previously undiscovered cell death pathways, such as necroptosis and ferroptosis, are present in low-grade, early-stage EEC. Finally, we have profiled (at the proteomic level) the immune microenvironment of these tumors, which is characterized by a high myeloid infiltration.

## Figures and Tables

**Figure 1 cancers-13-00794-f001:**
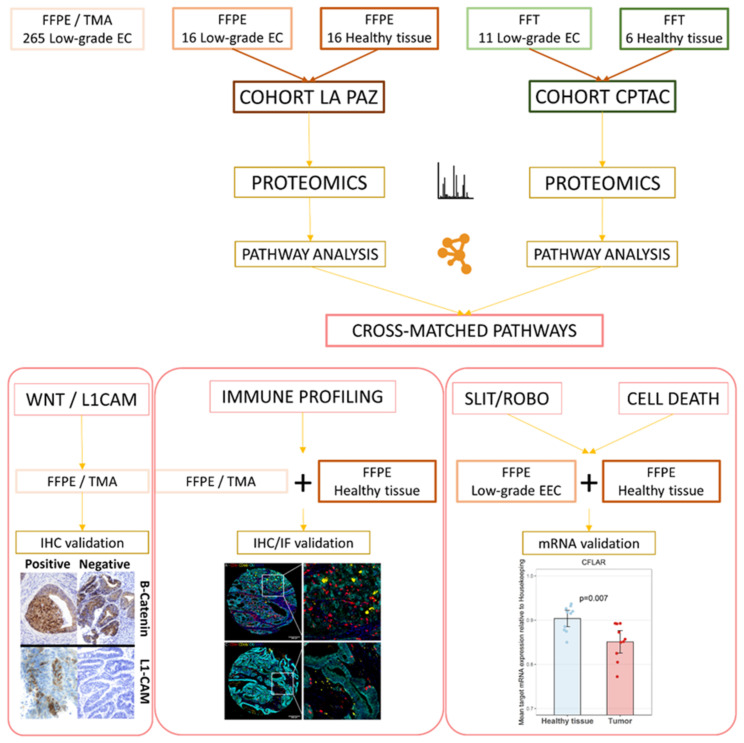
Experimental outline. FFPE: formalin-fixed, paraffin-embedded tissue. TMA: tissue microarray. EC: endometrial cancer. FFT: fresh frozen tissue. IHC: immunohistochemistry. IF: immunofluorescence.

**Figure 2 cancers-13-00794-f002:**
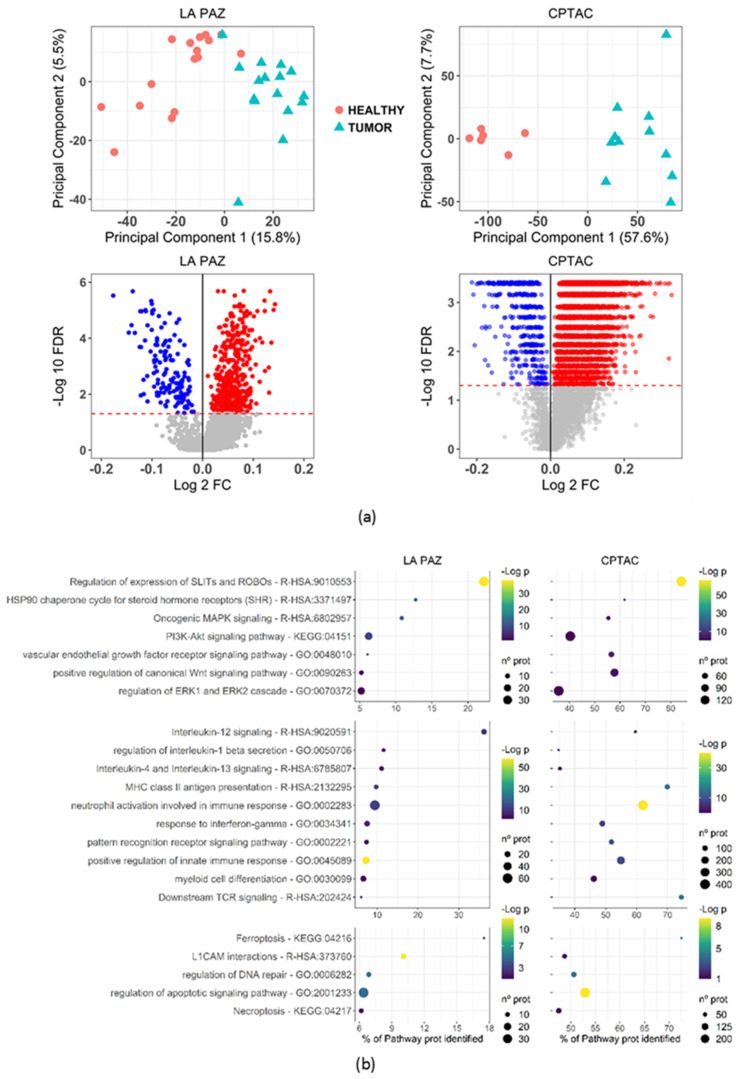
(**a**) PCA and volcano plots for the discovery and validation set. (**b**) Pathways identified in both cohorts segregated according to biological function (cell signaling, immune response or others). La Paz: discovery cohort. CPTAC: Clinical Proteomic Tumor Analysis Consortium, validation cohort.

**Figure 3 cancers-13-00794-f003:**
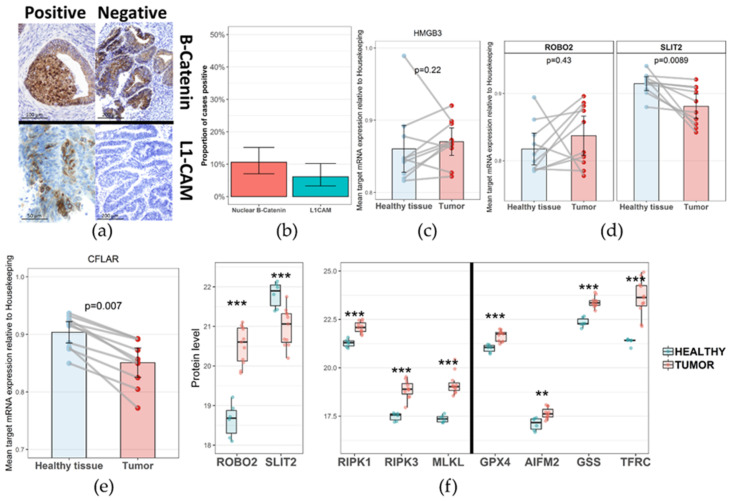
(**a**) Representative images of IHC evaluation. (**b**) Proportion of cases positive for nuclear β-catenin and L1CAM. Error bars denote 95% CI. (**c**–**e**) mRNA expression levels for relevant transcripts. The *y*-axis represents target mRNA expression relative to housekeeping. *n* = 10, two technical replicates. Bars represent mean and 95% CI. Grey lines connect paired tumor and healthy samples. (**f**) Relevant pathway member individual protein expression in the validation cohort. 11 tumor samples and 6 healthy tissue controls. IHC, immunohistochemistry. ** = *p* < 0.001, *** = *p* < 0.0001.

**Figure 4 cancers-13-00794-f004:**
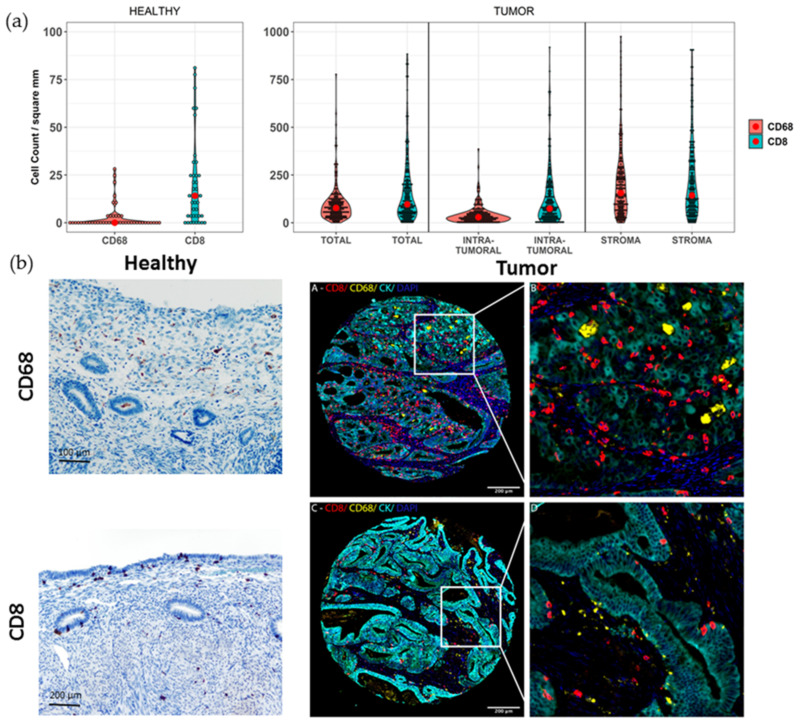
(**a**) Violin plots of cell density per spot. Healthy tissue graphs are scaled up to 100/mm^2^ and up to 1000/mm^2^ for tumoral tissue. Red spot denotes the median. (**b**) Representative microphotographs of immune infiltrates in healthy and tumor tissue. CD8, CD68 and Cytokeratin are represented by red, yellow, and cyan, respectively.

## Data Availability

The data presented in this study are openly available in ProteomeXchange at doi:10.6019/PXD022733, reference number: PXD022733.

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
