# Peer review of "Proteomic Analysis of Low-Grade, Early-Stage Endometrial Carcinoma Reveals New Dysregulated Pathways Associated with Cell Death and Cell Signaling"

_cancers, 2021, doi:10.3390/cancers13040794_

Round 1

Reviewer 1 Report

The manuscript from López-Janeiro et al. analyzed the proteome profiles for 16 FFPE tumor samples with paired healthy tissues from low-grade, early-stage EEC in the discovery phase and identified several dysregulated pathways based on the differentially expressed proteins. The results were further confirmed with the proteome data from CPTAC cohort with 7 healthy tissues and 10 tumor samples. Validations by using IHC staining and mRNA expression measurement were performed for selected proteins and pathways. The author identified some well-documented pathways such as WNT, AKT/Pi3K as well as newly reported SLIT/ROBO signaling, necroptosis, and ferroptosis. This work is important, focusing the potential pathogenesis in early-stage ECC. However, several issues should be further revised or addressed before publication. Following are comments and suggestions:

  1. Result: Because the “Materials and Methods” is arranged in the last section, so please add brief descriptions for the discovery and the validation cohorts in the beginning of “Result” in order to better understand the whole study design.
  2. Result: Regarding the validation cohort, the author mentioned that they downloaded the results from CPTAC data portal (data of accession, March 8, 2020). However, I can’t find this data of accession in that website. I think they might download the data from “Proteogenomic Characterization of Endometrial Carcinoma” (Cell, 2020). In this paper, 83 endometroid tumors and 49 adjacent normal tissues were analyzed, of which 64 tumors fit the recruitment criteria of Grade 1,2 and FIGO stage I, II. The author should include the proteome results from all these 64 tumors and 49 adjacent normal tissues, instead of collecting 7 controls and 10 patients. If this is not the paper the author analyzed, please add this one for more comprehensive analysis. In addition, it not clear whether the author downloaded the MS raw file and re-analyzed the proteome data by their own parameters or just directly adopted their proteome results. Please clarify it in the manuscript.
  3. The author mentioned the identification of 3112 and 9802 proteins in the discovery and validation cohorts, respectively, with a global FDR below 5% (page 3, line 95). But they described a FDR of 1% for peptide identification (page 10, line 346). Typically, a FDR less than 1% is allowed for identification by using orbitrap instruments. Please clarify whether 5% or 1% FDR is applied for database searching.
  4. Please prepare a supplementary table for the identifications in both discovery and validations sets to better compare the consistency between two proteome experiments.
  5. Figure 2: the author validated the nuclear β-catenin and membranous L1CAM proteins in 245 and 213 tumors by IHC, respectively. However, a very low percentage of positive results were found for β-catenin (10.6%) and L1CAM (6.1%), suggesting low detection sensitivity of these two proteins or the involved WNT signaling and L1CAM interaction in the early-stage ECC which is not consistent with the discovery in this manuscript. Please provide explanations.
  6. Figure 2 (c-e): please provide the mRNA expression levels of each proteins in individual tumor and normal sample and applied statistical analysis to evaluate whether the difference is significant. The protein expression levels in the discovery set should be included for comparison.
  7. Figure 2 (f): please add the protein expression levels of these selected proteins in the discovery set for better comparison between discovery and validation cohort
  8. Please confirm the settings for mass tolerance for MS1 precursor and MS2 fragment ions in Data Analysis (page 10, line 342-343). In current version, only the mass tolerance for MS2 is described.

Minor errors:

  1. Page 1, line 35: Please clarify EC as abbreviate for endometrial carcinoma throughout the manuscript.

Reviewer 2 Report

In general, this is a very well written manuscript and the topic is certainly of interest to the readership of Cancers. In this report, the authors investigated low-grade, early-stage endometrial carcinoma using quantitative mass spectrometry. Based on the subsequent gene ontology and pathway analysis, multiple dysregulated pathways/proteins were identified, which included SLIT/ROBO signaling pathway, necroptosis/ferroptosis and innate immune response.

While this report provided interesting insights in understanding this disease, it failed to provide sufficient evidence in correlating the identified proteins with disease diagnosis, classification, prognosis, treatment selection/development. The significance of this report is weakened in the case of no clinical relevance could be identified (partially due to the small sample size). In addition, one major issue for this report is the validation of the identified targets, more details and evidences (will explain in below) need to be provided to support the conclusion.

Here I have several more detailed comments regarding to this manuscript, which are listed based on page/line number:

Page 2, line 93: it would be clearer to the audience if the author could provide more details in the discovery and validation cohorts here (in addition to the Methods section), such as the criteria, sample size, etc. Also I noticed that for the discovery cohort, 16 FFPE tumor samples with paired healthy tissue were used, while for the validation cohort, 10 tumor + 7 healthy fresh frozen samples were used, are they also paired samples, please provide more details for the validation cohort. In addition, I am a little confused about the use of discovery and validation cohorts. Seems like here the authors were just comparing the result from two cohorts and looking for dysregulated targets in both cohorts? That is not the general way in using discovery and validation cohorts, please revise.

Page 4, figure 1: for figure 1a, please point out all the key proteins in the volcano plots, so that the audience would know where they are located. Also for all the key proteins, are they dysregulated across all the tumor samples? Please provide more details in the main text (eg. fold change, sample size, biological/technical replicates, etc) and in supplementary material (eg. the whole list of identified proteins and their quantification).

Page 4, line 118: here the authors used another cohort, containing 265 patients, to validate the finding related to β-catenin and L1CAM. Is there any reason to incorporate another cohort for validation? Please provide more details for this new cohort. It would be better if the author could provide IHC data for both discovery and validation cohorts, as it would deliver direct evidence in increased expression of β-catenin and L1CAM in tumors comparing with paired healthy samples. In addition, for the 265-patients cohort, why 245 samples measured for β-catenin and 213 for L1CAM? Is there any criteria in selecting samples? Is there any data available for paired healthy tissues, as it is not clear what is the expression level in healthy tissues. The low positive rate, 26/245 and 13/213, could not provide sufficient evidence in supporting the upregulation level of β-catenin and L1CAM in tumors.

Page 5, line 124: the authors jumped to measure the mRNA level of HMGB3, ROBO2, SLIT2, CFLAR for their discovery cohort, rather than the 265-patients cohort, please provide more explanations and rationales in using different cohort for validation. As reported in method 4.6, seems like only 10 paired samples were used? Also, please plot the mRNA level (ratio over gatekeeper gene) of healthy tissue and tumor samples separately for Figure 2c-e, not fold change, so that p-value could be calculated (please indicate whether there is any significant difference in the figures). And is there any reason why only two technical replicates were incorporated here? In general, triplicates is required to achieve confident measurement.

Page 5, line 136: the authors reported “individual protein measures” for several key proteins using their validation cohort? Again, why use a different cohort here for validation. What is the method used for measuring “individual protein measures”? There is no method reported for this measurement. If western blot was used, please provide the band and full membrane to support quantification in Figure 2f. And please indicate what is the y-axis for Figure 2f, and how the quantification was performed.

In general, it would be better if the authors could provide solid validation consistently using one cohort, and confirmed the expression level of all selected key proteins at both transcription and expression level.

Page 5, line 163: the authors incorporated 15 healthy uterine tissue to quantify CD8+ T cell and CD68+ myeloid cell populations, and compared the result derived from a cohort of 162 patients. Two issues here, according to method 4.7, there are in total 265 patients tissue available for TMA construction, and here only 162 was reported? Is there any criteria in selecting patients, and why only a partial of samples were analyzed? Second, seems like for healthy tissue, IHC was used for quantification, while for tumors IF was used instead? As the staining method is different (with different sensitivity and specificity), it is hard to compare cell population across different methods.

Page 6, Figure 2: please add scale bar for figure 2a. Please define asterisk for figure 2f. Please provide more details in figure caption for each experiment, such as sample size, biological/technical replicates, etc.

Page 6, discussion: the discussion involved many hypotheses, it would be better if the authors could use some cellular assays (knock-down/off or overexpression of some key proteins) to provide more evidence and solid support.

Page 7, Figure 3: please add scale bar for figure 2b. Please provide more details in figure caption for each experiment, such as sample size, staining method, etc.

Page 10, line 352: the dataset could not be found using identifier PXD022733, please double check the info is correct.

Please consolidate all the supplementary figures and tables into one signal files. Please also add cations for all supplementary figures.

Round 2

Reviewer 1 Report

The authors responded clearly to my questions. I think current version of manuscript is ready for publication.

Reviewer 2 Report

The authors carefully addressed some of the comments, but I still feel confused about the use of 3 different cohorts in this report.

First, for point 1, I went back to check the details about the CPTAC cohort (Cell 2020). As the authors stated, they incorporated their TMT number 2 and 5 for analysis. While based on what I found, there are in total 6 tumor samples and 3 paired adjacent tissue in each run, in that sense, there should be 12 tumor+6 healthy samples in total. Why the authors stated 10 tumor+7 healthy samples instead?

In addition, for point 2, in terms of the selection of protein targets for further validation, I still did not see a good flow/reasonable logics in the manuscript. For example, as the authors stated, they used the “discovery” cohort and the “validation” cohort for crosscheck, while on the other hand, the authors said in the “discovery cohort” only 2 key proteins (GSS and GPX4) could be identified. That is a sign to me that at least between the discovery” and “validation” cohort, there is some discrepancy and thus the proposed protein targets are questionable/with low confidence. Also, for the supplementary figures, I did not see β-catenin, L1CAM, HMGB3, etc., marked in any of the figures, how is their expression level in this two cohort, please display in the figure.

For point 3, I am still not fully convinced due to the low positive rate of β-catenin and L1CAM IHC result, 26/245 and 13/213, this could not provide sufficient evidence in supporting the hypothesis that WNT pathway and L1CAM interaction pathways are involved in early-stage endometrial carcinoma.

For point 4, the authors did not respond to the concern why 10 paired samples, not 16, were incorporated for mRNA analysis.

For point 6, although using the same primary antibody, the differential cell number between healthy and tumor samples could still potentially due to different specificity in secondary antibody and different sensitivity in detection method. It is usually not acceptable to compare across two different methods.

One major concern for this report is (still), the validation is done by jumping between different cohorts: IHC-265patients cohort, mRNA-discovery cohort, protein level (only via data analysis, not orthogonal method)-validation cohort, the result is not convincing enough to support the proposed mechanism. Please consider to provide at least a full set of measurement use the exact same set of samples for all the discussed protein (measure both mRNA and protein level (western blot or IHC). For instance, for the discovery set, the author could already identify differential mRNA level, it should be straightforward to get a differential protein level also the proposed mechanism is real. Even the sample size would be small, we should expect to see a differential sign.

Round 3

Reviewer 2 Report

I am glad that the authors identified the errors in using the CPTAC data, but the issue is, I still did not see any experimental evidence to address my concerns about the validation performed in this report. I would still recommend to provide at least a full set of measurement use the exact same set of samples for the key proteins identified in this manuscript, measure at both mRNA and protein level (orthogonal method, eg. western blot or IHC). Validation via jumping between different cohorts is not convincing enough to support the proposed mechanism.

In terms of the low positive rate of β-catenin and L1CAM for the IHC result, the authors proposed that is due to the localization of targeted protein, the sample size, the sensitivity of the assay itself, however, I am not fully convinced about this statement. For example, L1CAM does not have the issue for localization, while it is still having a very low positive rate 13/213. If it is really due to the sensitivity of the assay, then the IHC result could not be considered as a good validation to support the proposed mechanism. Also as I requested last time, I still did not see the expression level of L1CAM in the discovery and validation cohort (the authors only provided β-catenin and HMGB3 in figure S2). In terms of the sample size, the argument the authors had here indeed support my concern in using different cohort for validation. The validation should at least perform using the same set of samples and then applied to other cohort, ideally with larger size.

For the issue related to figure 4, I am glad that the authors agreed it is not appropriate to make direct comparison between different methodologies. However, I still see the comparison in main text (line 190), again, the higher infiltration rate in tumor samples could just due to higher sensitivity of the detection method. Please also point out in the main text that the staining methods for tumor and healthy samples are different.

Also according to the authors’ response towards point 6, seems like the authors first got the β-catenin and L1CAM result for the 265-patient cohort and then selected 16 cases to construct the “discovery” cohort for proteomic analysis? If that is the case, it is totally reversing the story descried in this manuscript, and I did not see anywhere mentioning that the discovery cohort is part of the 265-patient cohort? And if that is the case, the authors should have the IHC result for the discovery cohort, how is the expression level of β-catenin and L1CAM for this small set of samples, what is the positive rate?